# Mitigation of the Water Crisis in Sub-Saharan Africa: Construction of Delocalized Water Collection and Retention Systems

**Adolfo F. L. Baratta [1,\*], Laura Calcagnini [1], Abdoulaye Deyoko [2], Fabrizio Finucci [1], Antonio Magarò [3] and Massimo Mariani [3]**

1   Department of Architecture, Roma Tre University, 00153 Rome, Italy; laura.calcagnini@uniroma3.it (L.C.); fabrizio.finucci@uniroma3.it (F.F.)
2   École Supérieure D'ingénierie, D'architecture et D'urbanisme, 3228 Bamako, Mali; abdoulaye.deyoko@wanadoo.fr
3   Department of Architecture, University of Florence, 50121 Florence, Italy; antonio.magaro@unifi.it (A.M.); massimo.mariani@unifi.it (M.M.)
\*   Correspondence: adolfo.baratta@uniroma3.it

**Abstract:** This paper presents the results of a three-year research project aimed at addressing the issue of water shortage and retention/collection in drought-affected rural areas of Sub-Saharan Africa. The project consisted in the design, construction, and the upgrade of existing *barrages* near Kita, the regional capital of Kayes in Mali. The effort was led by the Department of Architecture of Roma Tre University in partnership with the Onlus Gente d'Africa (who handled the on-the-ground logistics), the Department of Architecture of the University of Florence and the École Supérieure d'Ingénierie, d'Architecture et d'Urbanisme of Bamako, Mali. The practical realization of the project was made possible by Romagna Acque Società delle Fonti Ltd., a water utility supplying drinking water in the Emilia-Romagna region (Italy) that provided the financing as well as the operational contribution of AES Architettura Emergenza Sviluppo, a nonprofit association operating in the depressed areas of the world. The completion of the research project resulted in the replenishment of reservoirs and renewed presence of water in the subsoil of the surrounding areas. Several economic activities such as fishing and rice cultivation have spawned from the availability of water. The monitoring of these results is still ongoing; however, it is already possible to assess some critical issues highlighted, especially with the progress of the COVID-19 pandemic in the research areas.

**Keywords:** water crisis in Africa; water collection and retention systems; sand dam; migration; climate change

## 1. Introduction

Mali is a landlocked country located in West Africa and 51% of its land is occupied by desert. The cultivated area is 4.7 million hectares, approximately 4% of the entire territory [1]. The country is characterized by the following major types of land:

- Mildly ferralitic soils [2], about 2 million hectares in the extreme south of Mali;
- Tropical iron-rich soils [3], over 17 million hectares in the southern area of the Sahel and south of Sudan. These are highly fertile soils;
- Arid soils in the same areas;
- Semi-arid soils, with a very dry climate, about 43 million hectares, corresponding to 35% of the territory;
- Hydromorphic soils [4] and vertosols [2], characterized by an excess of water due to a temporary or permanent clogging of the soil. This type of soil is dominant in depressions and basins, especially in the inner Niger delta, and it contributes to the alluvial behavior of seasonal streams.

About 47% of the Malian territory is made up of the Niger basin, while the basin of the smaller watercourse called the Senegal River covers 11% of the territory. The Volta basin, the third largest river, corresponds to 1% of the country's surface area, while the remaining 41% is covered by the Sahara Desert. Of note, 1700 km of the 4200 km of the Niger River run through Mali. The Niger and Senegal rivers, and the intricate network of tributaries, provide most of the permanent sources of surface waters. The total average surface water volume is estimated at around 50 km$^3$/year. Niger alone contributes 35 km$^3$/year, a third of which is wasted by evaporation. Renewable water resources, present in the subsoil, can be estimated at around 20 km$^3$/year, half of which is water in common between surface and subsoil. Therefore, in the whole country, the total volume of renewable water is equal to 60 km$^3$/year. The surface water resources entering the country amount to 40 km$^3$/year, mostly from New Guinea (33 km$^3$/year) and from the Ivory Coast (7 km$^3$/year) [5]. For the exploitation of these water resources, the country counts on the presence of five dams (Table 1), for a total water capacity of 13.8 km$^3$ (Figure 1) [6].

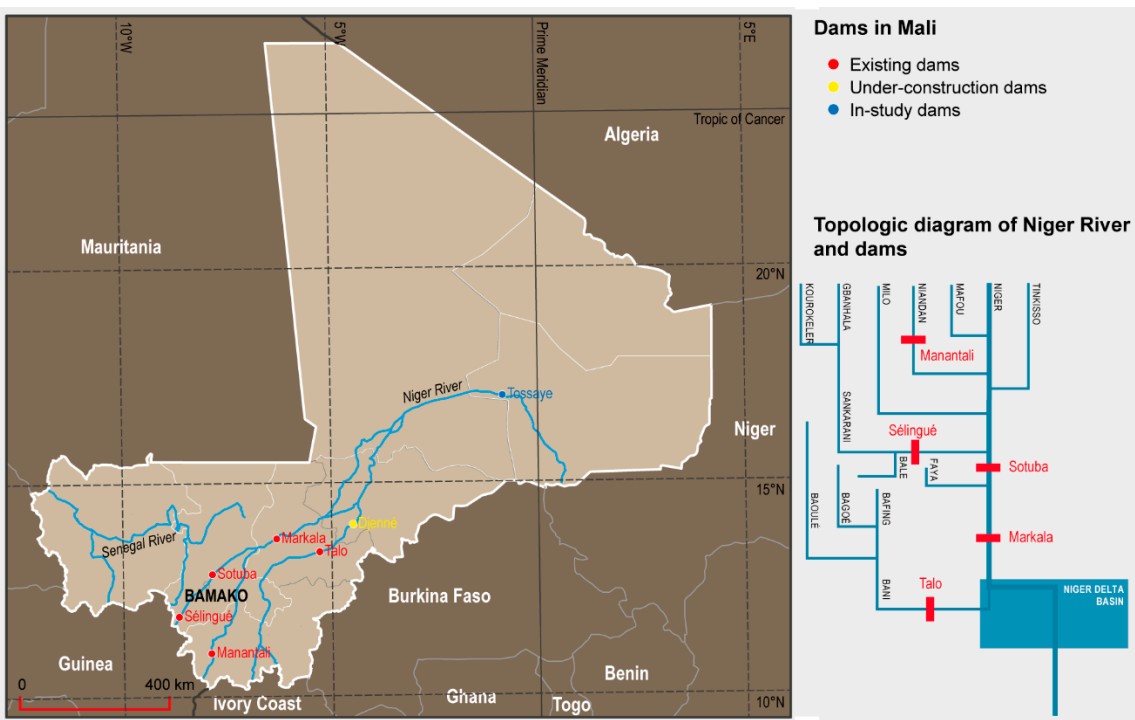

**Figure 1.** Water flows of the Niger River system and dams' position (Source: Authors').

**Table 1.** Description of the main dams on Mali's rivers [6].

| Dam | River | Capacity | Description |
|---|---|---|---|
| Sélingué | Sankarani | 2.17 km$^3$ | Produces hydroelectric power, controls the flow rate (reduced to 75 m$^3$/s at the level of Markala) and irrigates approximately 2000 hectares of land |
| Sotuba | Niger | 0.005 km$^3$ | Feeds a small hydroelectric power station and the Baguineda artificial canal, which irrigates 3000 hectares of land |
| Markala | Niger | 0.175 km$^3$ | Feeds a series of irrigation canals |
| Talo | Bani | 0.18 km$^3$ | Irrigates 20,000 hectares of land through controlled flooding |
| Manantali | Bafing | 11.27 km$^3$ | Prevents water shortages in the Senegal River during the dry season. Provides the reserve water for the irrigation of 15,000 hectares in Mali, 240,000 hectares in Senegal and 120,000 hectares in Mauritania and the production of electricity. |

Only 5% of renewable water resources available to Mali are exploited. Almost all of the water comes from seasonal surface sources, available only in the period from June to December. Only the withdrawals for the communities come from underground water resources, except for the city of Bamako, whose water is drawn from the Niger River [7]. The reasons why groundwater resources are so poorly exploited can be found in:

- Irregular feeding of the aquifers;
- Difficulties in locating subsoil water;
- High water withdrawal costs.

Furthermore, the complexity of the country's resource management organization hinders the implementation of any water policy (Table 2).

**Table 2.** Organization chart of the Ministries, Departments and Offices that deal directly or indirectly with water resource management [6].

| Ministry | Department/Office |
|---|---|
| Ministère de l'Agriculture (Ministry of Agriculture) | DNGR—Direction Nationale du Génie Rural (National Directorate of Rural Engineering), in charge of developing policies and strategies on hydro-agricultural management and services for rural communities; |
| | DNA—Direction Nationale de l'Agriculture (National Directorate of Agriculture), responsible for looking after green areas development and plant protection |
| | IER—Institut d'économie rurale (Institute of Rural Economy), in charge of economic studies and research on the agricultural sector |
| | CPS—Cellule de Planification et de Statistique (Planning and Statistics Unit), which deals with the collection and processing of data relating to agriculture |
| | Rural development offices, in charge of the management and development of medium and large-sized state-run irrigation grids |
| Ministère de l'Energie et de l'Eau (Ministry of Energy and Water) | DNH—Direction Nationale de l'Hydraulique (National Directorate of Hydraulics), responsible for national water policy, as well as for coordinating and monitoring its implementation, with powers over the inventory and management of resources |
| Ministère de l'Environnement et l'Assainissement (Ministry of Environment and Sanitation) | DNACPN—Direction Nationale de l'Assainissement, du Contrôle des Pollutions et des Nuisances (National Directorate of Sanitation, Pollution and Nuisance Control) |
| | AEDD—Direction Nationale des Eaux et Forêts, de l'Agence de l'Environnement et du Développement Durable (National Directorate of Water and Forests, of the Environment and Sustainable Development Agency), which intervenes in the event of environmental damage |

In addition, a new agency was created in 2002. The *Agence du Bassin du Fleuve Niger* (ABFN) (Niger River Basin Agency) was mandated with safeguarding the Niger basin, as well as the management and integration of water resources in coordination with the corresponding cross-border agencies. The intricate context of governing bodies has given rise to an equally complex regulatory framework through which the management of water resources enjoys little optimization. Ultimately, Mali is characterized by the presence of important water resources, but these are ill distributed over the territory and their strong seasonality is poorly managed.

The access rate to drinking water in Mali is 61% in rural areas and 69.2% in urbanized areas [8]. Bad distribution is followed by bad management, made up of a plethora of bodies with overlapping competencies, scarce powers and a complex regulatory system detached from the local realities (Table 3).

Besides its health implications, access to clean water is considered a human right and is a prerequisite for the realization of other human rights [9]. Therefore, the United Nations, with resolution 64/292 [10], has asked countries and international organizations "to provide financial resources, help capacity-building and technology transfer to help

developing countries to provide safe, clean, accessible and affordable drinking water and sanitation for all".

**Table 3.** Main regulations on water management in Mali [1].

| Year | Description |
|---|---|
| 1998 | National Environmental Protection Policy that aims to ensure a healthy environment and sustainable development, considering the environmental issue as central to every regulatory area |
| 2002 | Law number 02-006 of 31 January 2002, which updates the previous regulations on the subject and, in addition to declaring the public ownership of water, specifies the methods of management and protection, determining the rights and obligations of the State, local bodies and users |
| 2003 | National Wetland Policy that defines the long-term vision for the management of wetland ecosystems |
| 2006 | Strategic Framework for Growth and Poverty Reduction: a single legislative text that covers all medium-term development policies and strategies. It is the main document on which the negotiation with technical and financial partners is based. |
| 2006 | National Water Policy: provides strategic guidelines for the sustainable management of the country's water resources, respecting the balance between the land and aquatic ecosystems |
| 2006 | Law on Agricultural Orientation, which constitutes a unifying regulatory framework for all interventions concerning the agricultural sector |
| 2007 | National Health Policy and related sectoral strategies addressing solid waste from households and industry, the management of wastewater, special waste, and rainwater |
| 2007–2011 | Economic and Social Development Program that mainly concerns the agricultural sector, but, although indirectly, also the question of water resources |
| 2008 | National Irrigation Development Strategy which aims to standardize current approaches, identifying and highlighting the priority actions to be taken to make the most of the human and financial resources available. The strategy is based on the participatory and inclusive principle of the beneficiaries, who are involved in the definition, implementation, and management of irrigation projects. The strategy sets an increase rate of 9000 ha/year of additional irrigated areas: this goal was achieved every year from 2000 to 2010. |
| 2009 | Rice Cultivation Development Strategy, which aims to satisfy the internal consumption of cereals with the aim of transforming Mali into a net exporter of rice |
| 2010 | Code of State Property and Land, which includes groundwater and surface water, which are considered public property owned by the State |

The paper focuses on the research, design, and implementation process aimed at the construction and upgrade of existing *barrages* in rural areas of Sub-Saharan Africa.

*Water Crisis, Climate Change, Internal Conflict, and Migration*

Climate change is among the factors with the greatest impact on the water crisis throughout Sub-Saharan Africa. It will affect those countries, such as Mali, that heavily depend on more less diversified and strongly seasonal agriculture [11].

The climatic conditions of the country (Figure 2) are characterized by average temperatures between 27 and 30 °C annually, with large temperature variations occurring mainly in the desertic areas of the north [12]. In 2015, the maximum recorded temperature was 51 °C and the minimum was 10 °C [13]. The rainy season varies according to latitude: in the south of the country, it lasts up to 6 months, with a marked increase in rainfall between June and October, while in the north it is reduced to just three months between July and September.

The rainfall in the areas close to Sahara is only 50 mm/year, in the Sahel area it is between 100 and 1100 mm/year, while in southern Mali it exceeds 1100 mm/year [12]. Furthermore, Mali is in the so-called Intertropical Convergence Zone where the typical monsoons of West Africa occur. Due to climate change, between 1960 and 2015, the average temperatures increased by 1.2 °C, with a future expectation of linear growth: it is estimated that, by 2050, temperatures could increase between 0.9 and 1.5 °C, with the largest increase in the Kayes region. These changes could have an impact on the amount

and patterns of rainfall, the main source of hydrological supply. The rain cycle in Mali is decades-long and has undergone a decrease of 4.4 mm/year from 1950 to 1983 and an increase of only 2.6 mm/year between 1983 and 2015 (Figure 3). Mathematical models predict a slight average increase in precipitation (between 1% and 3%) together with a major decrease in the northern driest regions. Furthermore, the variation in the seasonal distribution could shift the wettest period towards the early part, between June and July with a subsequent reduction (between 6% and 10%) for the rest of the period (Figure 3). In addition, rain-related destructive phenomena are expected to increase [14].

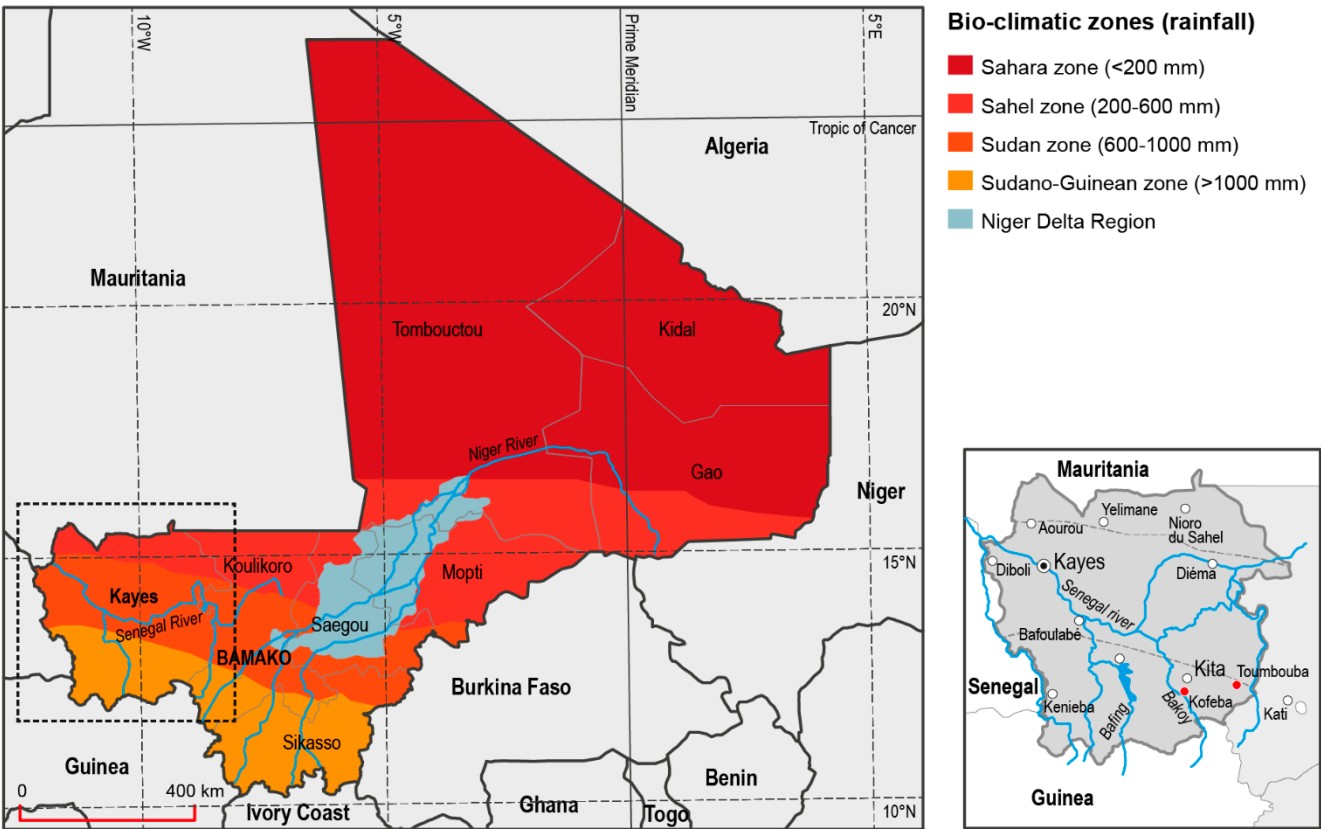

**Figure 2.** Mali bioclimatic zones and geographic classification of Kayes region (Source: Authors').

The impact on water resources could be substantial: there would be a reduction in the rate of subsoil resources and, at the same time, an increase in their need because the surface resources would tend to be less available due to intrinsic phenomena (such as the increase in evaporation because of the increase in temperature) and extrinsic (such as the growth in demand for water as a consequence of population growth) [15]. The result of climate change could be a decrease in food security (already under acceptable levels in the regions of Gao, Segou, Tomboctou, and Mopti) and malnutrition (acute childhood malnutrition already affects 13% of children under 5), resulting in increased mortality and reduced life expectancy [16].

The difficult political situation linked to internal conflict, which in June 2019 caused the forced migration of almost 148,000 Malians [17], could exacerbate this perspective. Political instability in Mali has its roots in the period following independence from France, obtained in 1960. The first independent government pursued real-socialism policies and nationalized all the industries in the country except for cotton. In the following five years, the country's economy nearly collapsed and Mali was forced to ask for financial support for currency to its former colonizer. A series of periods of crisis followed, leading to five coup attempts between 1970 and 1990. From the early nineties onwards, multiparty democracy began to consolidate, not without violence. Since 1990, the situation has become complicated

due to the revolts of the Tuareg people, who have settled in the north of the country. They gave life to the revolutionary movement *Mouvement Populaire de l'Azaouad* (MPA) with the aim of "liberating" the territory to the north by force [18]. The conflict officially ended in 1995 and re-exploded in 2007 due to the dissatisfaction of the Tuareg soldiers integrated into the Malian army. In 2012, the MPA was transformed into the National Movement for Liberation of Azawad (MNLA) [19], which found support in Islamic terrorist groups (Al Qaeda and Ansar). Moreover, Libya, until the fall of Gaddafi, supplied arms to the Tuareg that were superior to those available to the Malian army [20]. The declaration of independence of northern Mali, by the MNLA, saw the emergence of conflicting views between the Tuareg and Ansar. The MNLA and the Jihadist group clashed and brought the Tuareg closer to the national government [21]. However, in 2015 the Tuareg accused the Malian government of not respecting the agreements and the terrorist attacks began again.

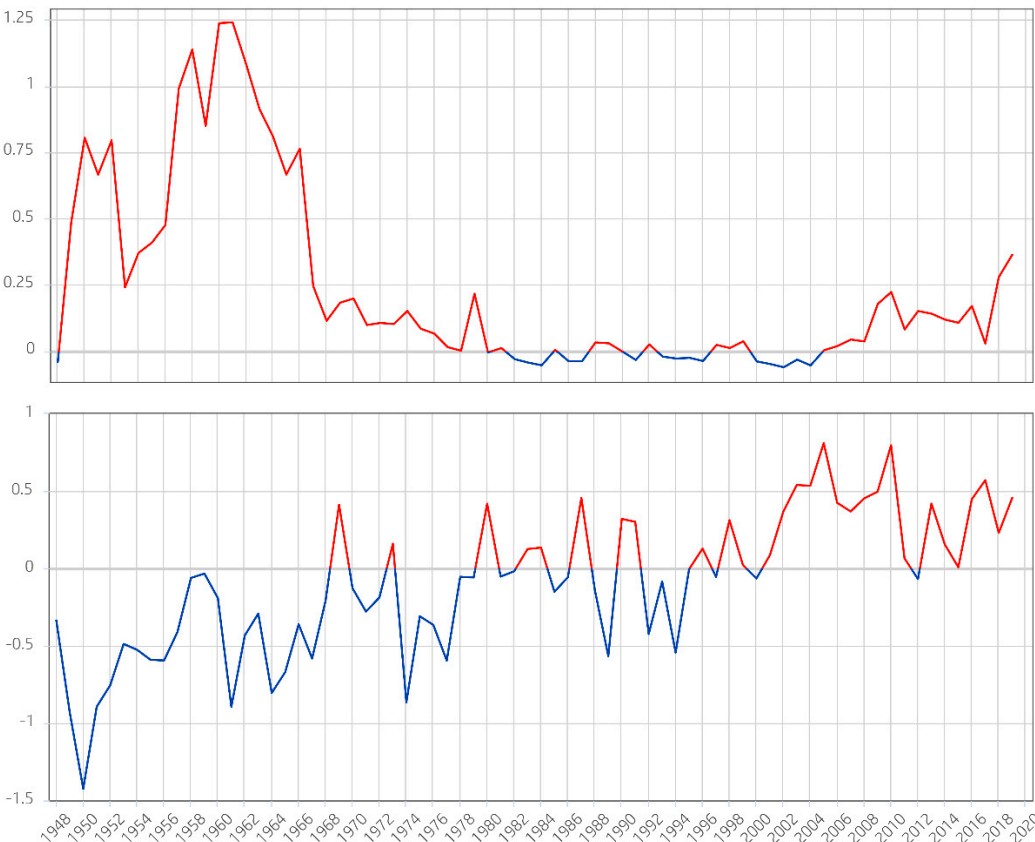

**Figure 3.** Anomaly of total precipitation [m] compared with anomaly of 2 m temperature (°C), historical series, in Sahel (12° N–17° N, 18° W–42° E). (Data source: Climate Change Institute, University of Maine, 2020).

Climate change, food insecurity, and social, economic, and political instability cause migration. Migration flows abroad come mainly from rural areas (73%), are characterized by a male majority (66%) and have as their primary destination the Ivory Coast (70%). As for long-distance destinations, due to the tightening of entry conditions in some European countries of traditional destinations such as France, the choices of Malian migrants fall on southern European countries such as Spain and Italy.

Rationalizing and improving the management of water resources, especially at the local level, would promote food safety, with the consequent reduction in mortality. In addition, improving the country's agropastoral economy would mean reducing political instability, and, over time, popular revolts. Therefore, a stable water supply would improve the living conditions in the country and reduce migratory phenomena.

## 2. Materials and Methods

This three-year research project (2017–2020) started with the cultural and scientific co-operation agreement between the nonprofit organization *Gente d'Africa* and the Department of Architecture of Roma Tre University (P.I., Prof. Adolfo F. L. Baratta, Research unit coordinator Prof. Fabrizio Finucci), with the aim of providing guidelines for the self-construction of infrastructure dedicated to health, food, and water in the depressed areas of Sub-Saharan Africa. Since 2015, the research group of Roma Tre University has been involved with addressing housing, social, and health problems in marginal areas of the world. With the aim of intervening specifically to address the water crisis in Mali, the two partners obtained the operational collaboration of the University of Florence and *Architettura Emergenza Sviluppo* (AES), a nonprofit organization founded in 2016. *Romagna Acque Società delle Fonti* Ltd. provided most of the financing. Furthermore, since 2019, an international cooperation program has been in place between the Department of Architecture of the University of Roma Tre and the *Ecole Superieure d'Ingénierie d'Architecture et d'Urbanisme* (ESIAU) of Bamako (P.I.: Prof. Abdoulaye Deyoko). ESIAU represents the only point of reference for architecture teaching in the entire country. This collaboration produces the planning, design, and construction of small and medium-sized infrastructures, with prevalent hydraulic characteristics in urban and rural areas. These water infrastructures include the distribution network, terminals, and water capture systems aimed at creating artificial basins, the so-called *barrages*. The *barrages* are barriers with a dam function, equipped with locks for the control of flooding.

The project was carried out in the following phases:

1.  Analysis: identification of the architectural, technological, structural, and environmental issues influencing the design choices; adoption of appropriate survey methodologies with the aim of identifying morphological, functional, and technological solutions for the selection of suitable construction techniques; cost-benefit analysis of the solutions identified.
2.  Design proposal and execution of the works: the university teams dealt with the final projects design, from the elements and components design up to the complete construction. In this phase, students, researchers, and professors were involved, with the aim of theoretical and practical cultural exchange. The nonprofit organizations took care of the institutional relations and logistical aspects, while the financing partner supported the proposition activity through its know-how.
3.  Collection and dissemination of results. The nonprofit organizations, in particular AES, took care of the promotion and communication of the initiative which was presented at an International Conference held on 15 September 2018 at the Department of Architecture of the Roma Tre University and exhibited at EXCO 2019, an event held in Rome and dedicated to cooperation and development of distressed areas.
4.  Monitoring of results, currently in progress.

The Italian and Malian universities have focused on the opening of the Erasmus Program beyond the European borders, implemented by the European Commission through the Key Action 107, International Credit Mobility, responding to the Call 2019 and being recently (August 2020) awarded the funding for carrying out the activities of cultural cooperation and international educational exchange.

Although the French term *barrages* is mainly used to indicate medium and large dams, in the context of rural settlements it is used to allude to small structures, able to stop or channel the waters coming from the swelling of the streams during the rainy season. In Africa, such structures, also known as sand dams, help the replenishment of the aquifers [22]. *Barrages* are containment structures that can be classified according to the ability to convey subsoil water or surface water into two types (Figure 4):

- Underground *barrages*;
- Surface *barrages*.

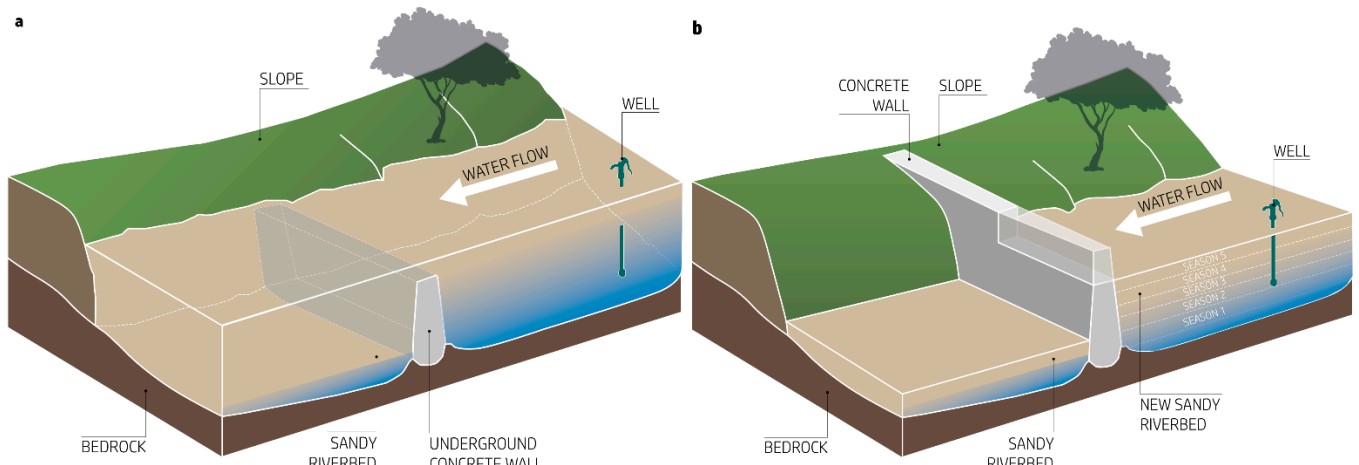

**Figure 4.** Concept diagram of *barrage* (sand dam): (**a**) Underground *barrages*. (**b**) Surface *barrage* (Source: Authors').

The underground *barrages* are made by digging a deep excavation down to a rocky layer or sufficiently compact soil, allowing for an ordinary foundation of limited size and preventing water from penetrating to greater depths, determining the depth of the aquifer.

The surface *barrages* follow the same principle; however, they are set on a shallow foundation and are designed to stop the watercourse when it swells on the surface because of abundant rainfall on geologically impermeable and compact clay soils.

The *barrage* structure works in a simple but effective way with minimum environmental impact. Because of its small size, it allows overtops water to flow over the *barrage* and avoids draining the aquifer downstream, slowing down the flow of water and facilitating the filling of wells [23].

The surface *barrages* generate an artificial basin which, fed during the rainy season, forms a water reservoir ahead of the dry season. Both *barrage* types, when locally built, use simple construction techniques and intuitive technical principles based on locally available materials. As a result, most of the existing surface *barrages* in Mali are weak structures, often without any foundation. They are incapable of resisting the water pressure as they exploit their weight and shape instead of the characteristics of the material. The poor culture of maintenance means that they are destined to become partially buried due to the accumulation of debris on the upstream-facing side. Furthermore, the lack of a system of local know-how transfer prevents the diffusion of the construction techniques, especially when not based on traditional methods like the use of raw earth, and the correct propagation of a "rule of the art" that could consolidate over the time and could spur any innovations.

However, in many cases the surface and underground *barrages* have been combined in a single structure capable of slowing down the surface flow and increasing the infiltration of water into the subsoil, ensuring greater availability of water resources.

## 3. Results

One of the main results of this project was the reconstruction and monitoring of two *barrages*, respectively near the village of Toumbouba and the village of Kofeba, close to Kita, one of the most important towns of the Kayes region, 200 km from Bamako. Both are surface *barrages* although they are different in size and required a completely different approach.

### 3.1. The Barrage in Toumbouba

Located at the geographical coordinates 13°01′07.58″ North Latitude and 9°21′01.04″ West Longitude, the Toumbouba *barrage* (locally known as Tumumba) is grafted onto a preexisting structure built a few decades ago. The original *barrage* was a stone masonry structure, 60 m long, grafted to the north in a retaining wall of an embankment, orthogonal to the position of the main structure. The containing wall resisted the horizontal thrust

due to its shape, having a triangular section with the hypotenuse inclined by about 45°. The structure was built on ordinary foundation made by an excavation of a few tens of centimeters below the ground level, filled with large aggregate material bound using cement. The elevation structure had a larger size than the foundation, laying at least two thirds directly on the riverbed. The historical memory of the village traces its construction back to the mid-eighties, but the oral tradition did not confirm any certain source.

During the first on-site survey, in April 2017, it was determined that the power of the watercourse in 2007 caused the overturning of the structure and its breaking into three sections. The first section remained firmly buried for its entire length, while the most stressed section, about 15 m long, divided into two parts, laying down on the bed of the watercourse showing the weakness of the construction technique. The torsion due to the overturning caused damage in need of repair to the orthogonal retaining wall, and the erosion of the water caused the dissolution of a large part of the embankment it contained, jeopardizing the necessary shape of the reservoir. During the survey, the research team acquired the preliminary knowledge of the availability of materials and the skills of the local workforce. It learned of the almost absent theoretical and practical knowledge of the use of steel for construction of flexible structures, and, on the other hand, of the recent use of concrete blocks. These could be easily made in large quantities with handmade block molds as demonstrated in the large urbanizations of Bamako and Kita, where they have replaced the use of the traditional Malian raw earth adobe techniques. The survey ended with the inspection of the damaged structure, to proceed with the design phase. The mold for the construction of the concrete blocks for the vertical elevation was designed first, while the two overturned sections were demolished, and the rubble removed. Those operations were carried out with the help of nearby villagers. The design of an armed and buttressed structure was prepared next, one able to offer the greatest resistance to horizontal thrust, combining shape and flexibility. The construction costs were summarized in a bill of quantities, using the prices recorded in the initial inspection phase as a reference and cross-checking them with the international price list integrated in the CYPE software package [24]. The second mission, aimed at the construction, was organized into two phases. During the first phase, in February 2018, the base of the *barrage* was built. While waiting for the setting and curing of the concrete, the concrete blocks were manufactured. During the second phase, in April 2018, the *barrage* was built (Figure 5), and the gaps in the retaining wall of the embankment and on the embankment itself filled.

The result is a wall structure with variable thickness, strongly reinforced in both directions, buttressed, 15 m long and 2 m high, connected to the preexisting structure through the insertion of steel bars $\Phi16$ and concrete injections every 40 cm in elevation for a depth of at least 60 cm. The structure was finished with a 3 cm thick cement-based plaster, to account for the expected lack of maintenance in the future (Figure 6).

During the survey it was decided to intervene with the consolidation of the structure and the reinforcement of the embankment adjacent to the *barrage*. Furthermore, the correct hydraulic functioning of the structure was tested and proved capable of creating a water reservoir of 6.8 hectares with a depth varying between 30 cm and 2 m, corresponding to a water reservoir of about 45 thousand m$^3$ in the period of maximum capacity. The basin serves, directly or indirectly, a very vast territory comprising about 20 villages, or 40 thousand inhabitants.

### 3.2. The Barrage in Kofeba

The work for the Kofeba *barrage* benefited from the experience gained by the group during the work in Toumbouba. The introduction of several practical innovations in the workflow resulted in a shortened construction time without compromising full functionality.

Identified by the geographic coordinates 13°01′01.77″ North Latitude and 9°35′43.60″ West Longitude, the *barrage* is located near a village which in the maps takes the name of Forongo, locally known as Kofeba, located 15 km west of Kita. Like the *barrage* in Toumbouba, the *barrage* in Kofeba was a preexisting structure in stone laid with cement,

215 m long and 80 cm high, with a particular shape whose triangular section provides an acute angle of just 30° near the shutter plane. The regulation of the capacity of the basin was carried out through the manual handling of two rudimentary iron closures, hinged as doors without the presence of a linear counter frame or fixed frame.

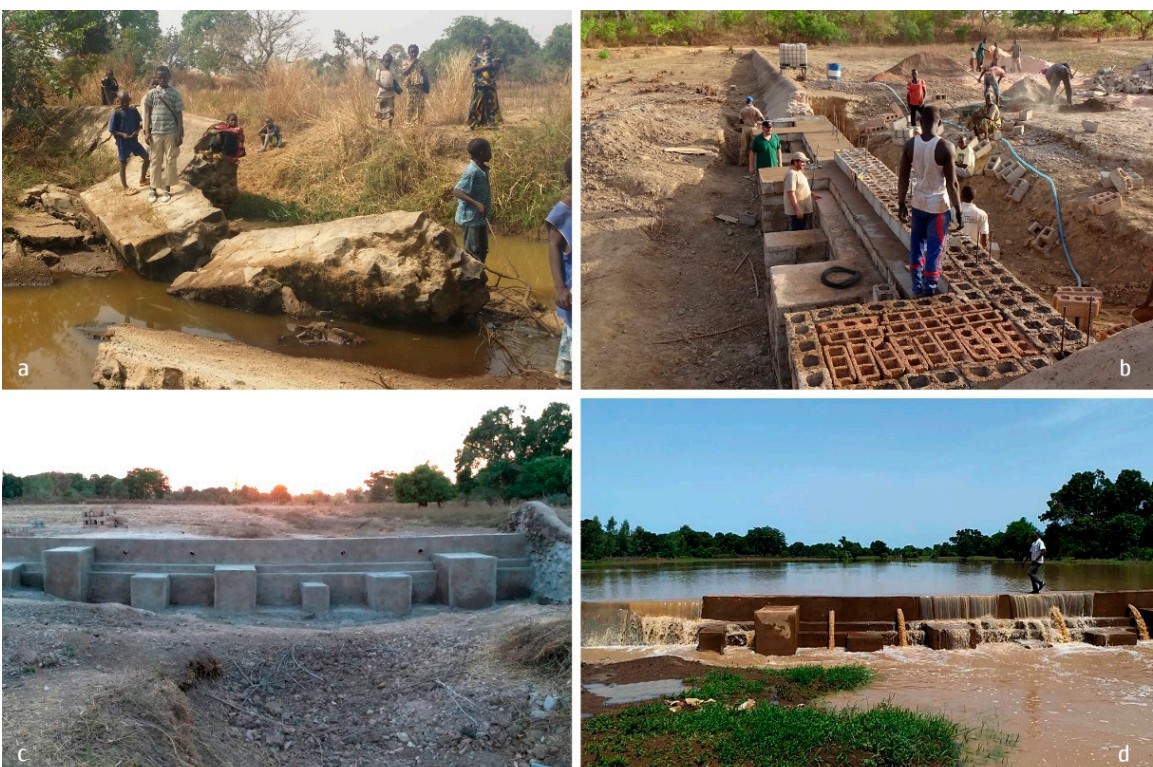

**Figure 5.** The rebuilding process in Toumbouba: (**a**) the old *barrage* down; (**b**) the building phase; (**c**) the wall complete; (**d**) the *barrage* working phase (Source: Authors').

Over time, this simple moving mechanism has yielded, bending from the rotation plane of the hinges due to the constant horizontal thrust. Because of both the design weakness of this system and the difficulty in finding suitable metal artifacts, it was decided to fill the openings with a wall. Like for the *barrage* in Toumbouba, some demolition works of part of the structure became necessary, because between the two doors there was a wall fragment about 1 m wide.

The demolition left a gap of about 3 m to be filled with the new wall. Given the small size, steps were taken with the aim of making the construction process more efficient and avoid an additional trip. Following a first mission carried out in December 2019, it was possible to ascertain the presence of a solid foundation structure, as wide as the thickness of the base of the wall, or about 2 m, and about 50 cm deep. Therefore, it was decided to proceed with the drilling of holes for the insertion of steel bars ahead of the reinforcement of the structure in elevation, filling the holes with concrete injections. The new wall was built with two imbedded water outlets to control water overflow.

While the work on the Toumbouba *barrage* was carried out without the aid of machinery or construction equipment, for the Kofeba *barrage* it was necessary to use a hammer drill capable of realizing the appropriate depth holes, powered by an electric power generator. In the lapse of time between the two missions, concrete blocks were built with the same procedure already used in Toumbouba, and the material necessary for construction was set aside.

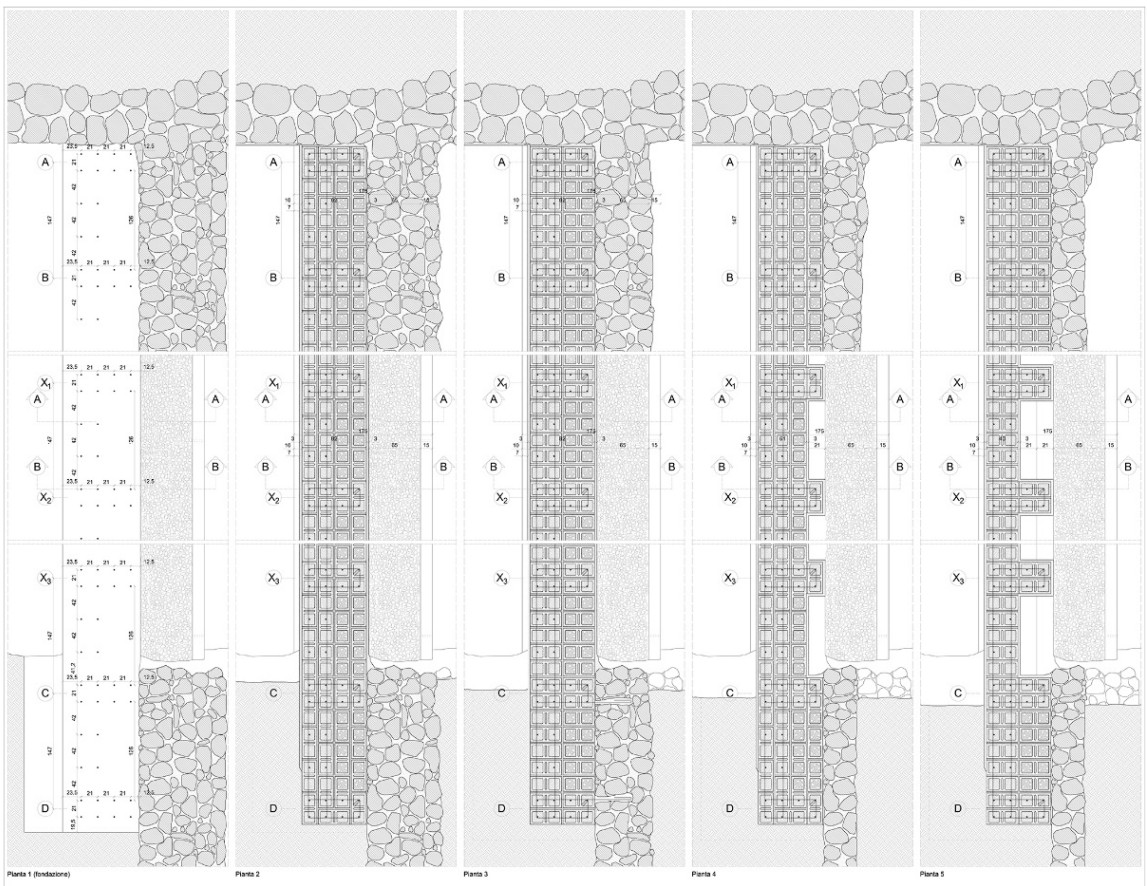

**Figure 6.** The executive design of the *barrage* in Toumbouba (Source: Authors').

The experience previously gained allowed the completion of the construction phase in only 4 working days (Figure 7). The monitoring phase of the Kofeba *barrage* has just started, since, at the time of writing, the rainy season had just begun, but the first photographic surveys to verify the strength of the structure appear satisfactory. At full capacity, in the absence of further settling of the reservoir grounds and with the help of small additional earthworks, the Kofeba basin could cover an area between 9.5 and 20 hectares.

Although belonging to the same typology, the two *barrages* have very different dimensional characteristics (Table 4).

**Table 4.** Comparison between the main data of the two *barrages*.

| | *Barrage* **of Toumbouba** | *Barrage* **of Kofeba** |
|---|---|---|
| Dimensional characteristics of the entire structure [m] | Length: 60 Height: 2 | Length: 215 Height: 0.8 |
| Extension of the body of water [ha] | 6.5–8.5 | 9.5–20 |
| Depth of the basin [m] | Min 0.3–Max 2 | Min 0–Max 0.7 |
| Amount of water [m³] | Min 45,000–Max 60,000 | Min 90,000–Max 140,000 |
| Number of beneficiary villages [units] | 20 | 35 |
| Number of beneficiary inhabitants [units] | 40,000 | 70,000 |
| Construction time [days] | 20 | 7 |

The Toumbouba *barrage*, although smaller in size, required considerable effort to restore the reinforced concrete foundation along the entire collapsed section. In addition, the water on the Toumbouba *barrage* reaches a very high speed, generating problems of resistance to dynamic thrust. For this reason, both the foundations and the wall in elevations required a greater mass and a longer construction time.

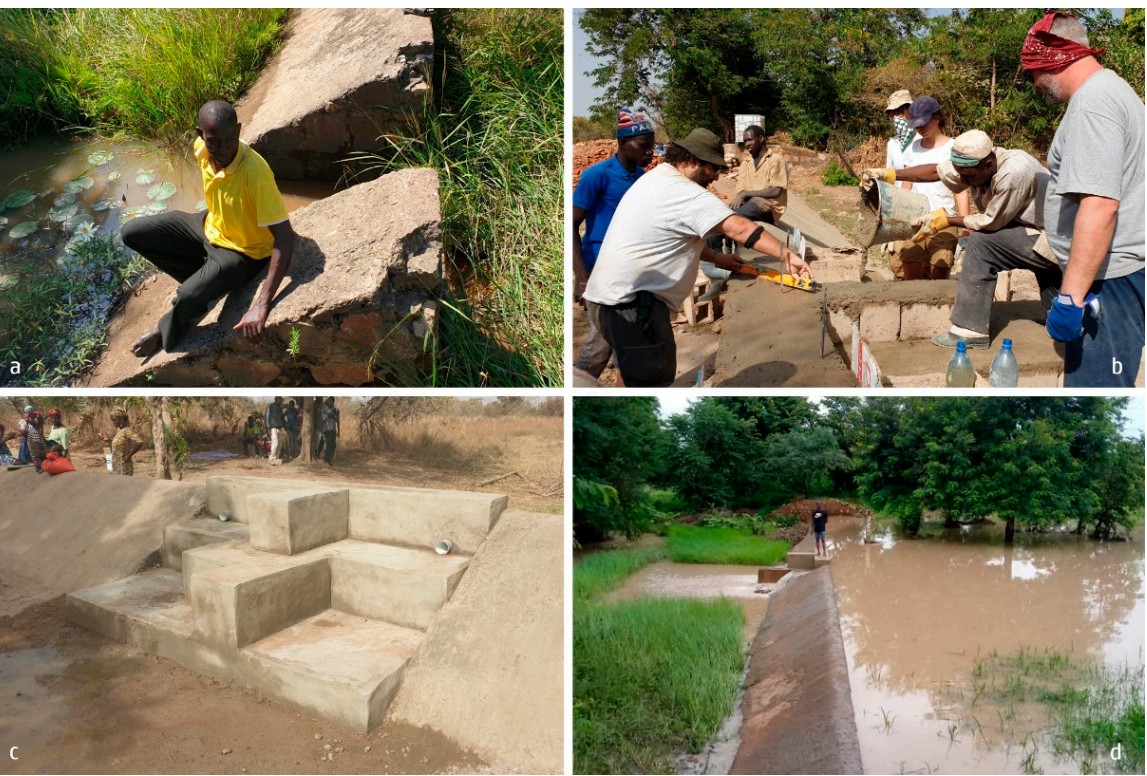

**Figure 7.** The rebuilding process in Kofeba: (**a**) the old *barrage*; (**b**) the building phase; (**c**) the wall complete; (**d**) the *barrage* working phase (Source: Authors').

Kofeba's intervention (although it is a larger *barrage*) is more contained in terms of size, moreover, it was possible to use the existing foundation section. The Kofeba basin tends to fill gradually with respect to that of Toumbouba and with a lower height of the water level. These aspects have allowed a smaller wall thickness and, therefore, a much shorter period of work. At the same time, however, the Kofeba area required numerous smaller restoration interventions of the natural embankment, carried out with boulders and concrete by the local labor force.

Both interventions on the *barrage*, for the number of villages and beneficiaries' inhabitants, are characterized by an excellent cost–benefit ratio.

## 4. Discussion

The reconstruction of the two small *barrages* has yielded several benefits that can be classified into two categories: direct benefits and indirect benefits.

Regarding direct benefits, the following can be mentioned:

- Reduction of water shortages during the dry season in a vast territorial area including dozens of villages and thousands of people. Replenishing the aquifer had an impact on the availability of water in wells even tens of kilometers away;
- Improvement of food supply, thanks to the possibility of growing rice crops in excess to the amounts needed for local consumption and the renewed presence of fish proteins. Furthermore, the nutritional diversity has been increased by the higher productivity of the gardens that can enjoy irrigation for more than one season and by the improved quality of the meat of the animals raised, especially poultry and sheep that can be adequately fed;
- Birth of small private business activities such as, for example, a small fish farm;
- Improvement of environmental well-being conditions, due to the mitigation of temperatures in the microclimatic environment surrounding the Toumbouba *barrage*, thanks to the presence of a mass of water capable of activating thermal inertia mechanisms.

About indirect benefits, the following should be noted:

- Improvement of internal social cohesion, due to the principle of sharing food resources derived from the creation of the water basin;
- Improvement of relations with neighboring communities, through the consolidation of existing exchanges, the activation of new relationships, and the presence of the conditions for a better territorial cohabitation, thanks to the reduction of conflicts;
- Improvement of economic prospects, through the possibility of diversifying income-generating activities due to the emergence of new needs. This mechanism can produce collateral micro-entrepreneurial activities to those related to the presence of water, improving the internal economy, and providing the conditions for improving living conditions;
- Transfer of knowledge to the local parties, both scientific partners and the general community;
- Improvement of the social function of the *barrages*. Thanks to the new shape of the *barrages* with terraced seats, people walk and stay on them.

Given the quality of the results achieved when compared to the very low costs of both interventions (less than 25 thousand euros), the benefits far exceeded the costs incurred. In particular, the benefits can be classified as:

- Environmental benefits, such as replenishment of groundwater reserves and retention of surface water reserves for the dry season; presence of water in the wells of numerous villages near the *barrage*; regulation of the microclimate in the hottest period thanks to the presence of water; improvement of the conditions of farmed animals;
- Economic benefits, such as the flourishing of various common business activities, directly related to water (such as rice cultivation, fish farming) or indirectly induced by the additional disposable income (tailoring, sale of meat and eggs, etc.);
- Social benefits such as improvement of the living conditions of the inhabitants; improvement of relations between inhabitants and social cohesion; improvement of food quality; activation of periodic parties; reduction of migrations of young people from the village to the capital; improvement of the quality of food, such as meat and locally produced vegetables.

The monitoring phase of the project is still underway to assess its full environmental and economic benefits. A feasibility study is currently looking at the construction of a control station for the acquisition and monitoring of environmental variables, such as temperature, humidity, and pressure, with the aim of demonstrating the micro-climatic variations due to the presence of the water basin.

Data collection relating to the economic benefits indirectly related to the presence of the water basins is ongoing. Several micro-entrepreneurial activities are being observed, such as commercial fish farming, rice cultivation for the sustenance of the community and for retail sale, the appearance of several vegetable gardens, and the introduction of the seasonal rotation of crops.

In particular, in the fish farm in Toumbouba, an artificial pond of about 150 m$^3$ in capacity was built. This pond was built by excavating; the walls were covered with reinforced concrete, while on the bottom a layer of about 20 cm of compacted earth was created, containing no less than 10% clay, measured by empirical sedimentation tests. This waterproof system, often used for depths not exceeding 3 m, is not recommended for greater depths, where it is advisable to proportionally increase the thickness of the compacted earth. In addition, depths that exceed the height of the aquifer should not be waterproofed in this way to facilitate filtrating of water. The tank requires pumping for the loading and drainage of water, which is periodically necessary for the removal of organic sediments. This periodic maintenance avoids regulating the concentration of oxygen in the water by means of additional mechanical systems. The water of the farm is derived from the *barrage*, by means of mechanical pumping. Therefore, a discontinuous flow was preferred, necessary to replace the evaporated water. To begin breeding, fifty pairs of a

local catfish species weighing between 300 and 400 g were placed in the tank. To facilitate reproduction, artificial nests were inserted into the tank made on the bottom with ballasted polyethylene pipes, 50 cm long with a diameter of 20 cm. Fishes were fed on alternate day manual shedding commercial feed. Despite an insignificant mortality, the fish are sold when they reach a weight between 2 and 2.5 kg, allowing to control the density in the tank. A high density can cause that bigger fish become predator of the others. The small entrepreneurial reality of the fish farm has allowed to create three new jobs in the village.

Favored by a rural economy still largely based on the bartering of goods and services, a series of artisanal activities were born in the villages with the aim of having fish in exchange. Three tailoring and fabric processing activities were born.

The production of concrete blocks for the construction of the *barrage* has allowed the specialization of some inhabitants. Two of them, in particular, have started production (through manual molding) and sale of the blocks to other villages for a total of 10,000–12,000 blocks per year.

Some of these activities are still very fragile and highly dependent on the stability of the retail market. Unfortunately, the recent unpredictable pandemic events have led to a significant reduction in the demand for fish farm products in neighboring markets, leading to the loss of fish stock already in the tanks. The pandemic has highlighted the fragility of entrepreneurial activities undertaken in these contexts that have to face a much higher entrepreneurial risk than elsewhere.

## 5. Conclusions

The water crisis in Sub-Saharan African countries, and specifically in Mali, is not a new phenomenon but in recent years has taken on characteristics with geopolitical and international implications. The ongoing climate changes highlight the dramatic consequences of water scarcity: absence of health and hygiene, malnutrition especially in children, with increased mortality rates in under-fives, and the constant decrease in arable land. These factors increase political instability and determine the incessant escalation of revolts, and popular uprisings often linked to international terrorist groups. In these conditions, migratory flows can only intensify, expanding the scope of the phenomenon and establishing the need for international intervention.

To solve the water issue it is necessary to promote the self-construction of small local infrastructures such as *barrages* and to simplify the unnecessarily complex and articulated state management and regulatory systems.

The three-year research project led by the Department of Architecture of Roma Tre University, in collaboration with the University of Florence and in partnership with the Onlus *Gente d'Africa*, *Romagna Acque Società delle Fonti* Ltd. and AES *Architettura Emergenza e Sviluppo*, has led to the reconstruction of two damaged and no longer operating *barrages*, in the Kayes region, near Kita.

The operational phase of the project saw the participation of local communities and the involvement of municipal authorities brought together to restore the functionality and increase the performance of two small and damaged infrastructures.

The research project encompassed a series of on-site missions involving information gathering, surveying, prospecting, design, manufacturing, construction and the collection and dissemination of results. The reconstruction of the basins produced several indirect benefits. These range from improved cohesion of local communities to the increase in micro-business activities related to the presence of water. In the current phase, the monitoring of the results is mainly aimed at assessing the environmental impact of the *barrages*, through the direct measurement of a series of basic environmental parameters and the indirect detection of the improvement of the living conditions of humans and animals.

**Author Contributions:** All authors wrote the paper. All authors have read and agreed to the published version of the manuscript.

**Funding:** The research was supported by Romagna Acque Società delle Fonti Ltd., Gente d'Africa onlus, Roma Tre University, AES Architettura Emergenza Sviluppo.

**Institutional Review Board Statement:** Not applicable.

**Informed Consent Statement:** Not applicable.

**Data Availability Statement:** Not applicable.

**Acknowledgments:** A heartfelt thanks to Sara Negroni, Mauro Foli, and Massimo Mantuano from *Gente d'Africa onlus,* And to Tonino Bernabè for *Romagna Acque Società delle Fonti.* A further heartfelt thanks goes to Diawoye Tounkara, local reference for *Gente D'Africa* but, above all, irreplaceable guide, and support in Malian territory. Finally, a special thanks to the students who took part in the various missions in Mali: Giovanni Baratta, Iacopo D'Orazi, Francesca Limongelli and Pietro Marinari.

**Conflicts of Interest:** The authors declare no conflict of interest.

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
