# Peer review of "Mitigation of the Water Crisis in Sub-Saharan Africa: Construction of Delocalized Water Collection and Retention Systems"

_sustainability, doi:10.3390/su13041673_

Round 1

Reviewer 1 Report

Line 3 wild waters regimentation…. (at least drop “wild”) better to find

another expression.

Line 54 km3/year > km3/year

Line 60 on Mali's waterways >> on Mali's rivers

Table 1 (a map with the locations can be usefull to the reader)

Markala: it serves to swell the river to feed >> check dam to feed a series

Talo: from this dam comes the water >> This artificial lake supply the water for

the irrigation of 20,000 hectares…..

Manantali: check reservoir for any…

Table 2 DNGR – _Direction Nationale du Génie Rural (National Di-rectorate

of Rural Engineer-ing), in charge of developing policies and strategies on hy-droagricultural management and services for rural communities;

It is repeated in the table under “structure” drop it

Ministère de l’Agriculture (Minister of Agriculture)

Lines 203, 208, 214, 219 underline the phases to help the reader

Lines 239-242 (poor technical English) reorder and explain in a more

technical way…..

Line 244 exceeds > overtops

Lines 250 – 256 > (poor technical English) rewrite in a technical way : it is not

clear and in some points even not correct: i.e. “able of resisting the water

pressure only by shape and not by material”

Line 270 the shoe wall (it is not technical language!)

Line 274 welded using a cement conglomerate (it is not technical language!)

The shutter of the elevated structure (it is not technical language!)

Lines 277-285 reorder and rewrite in a good English

The paper is very interesting describe some practical activity performed by the authors on site but the technical description is very poor. English review is needed.

Author Response

The authors thank the reviewer for the careful review and for the suggestions received for the improvement of the paper. An extensive revision of the English language and style was carried out.

The responses to the review points are shown below:

Line 3 wild waters regimentation…. (at least drop “wild”) better to find another expression.

The title of the paper has been revised.

Line 60 on Mali's waterways >> on Mali's rivers.

Text modified as suggested by the reviewer (new line 61)

Table 1 (a map with the locations can be usefull to the reader)                .

A map has been inserted as suggested by the reviewer.

Markala: it serves to swell the river to feed >> check dam to feed a series.

The full text in the table has been revised.

Talo: from this dam comes the water >> This artificial lake supply the water for the irrigation of 20,000 hectares

The full text in the table has been revised.

Manantali: check reservoir for any

The full text in the table has been revised.

Table 2 Direction Nationale du Génie Rural (National Di-rectorate of Rural Engineer-ing), in charge of developing policies and strategies on hy-droagricultural management and services for rural communities; It is repeated in the table under “structure” drop it

Ministère de l’Agriculture (Minister of Agriculture)

The duplicated part has been deleted.

Lines 203, 208, 214, 219 underline the phases to help the reader

The phases have been underlined as suggested by the reviewer (Line 195-200-206-211)

Lines 239-242 (poor technical English) reorder and explain in a more technical way…..

The sentence has been reordered by replacing some terms (Lines 228-230)

Line 244 exceeds > overtops

The word was changed as suggested by the reviewer (line 232)

Lines 250 – 256  (poor technical English) rewrite in a technical way : it is not clear and in some points even not correct: i.e. “able of resisting the water pressure only by shape and not by material”

The whole sentence has been revised (238-246)

Line 270 the shoe wall (it is not technical language!)

The term was replaced with “containing wall” (line 267).

Line 274 welded using a cement conglomerate (it is not technical language!)

The term was replaced with “cement” (line 267)

The shutter of the elevated structure (it is not technical language!)

The sentence has been revised (lines 267-270)

Lines 277-285     reorder and rewrite in a good English

The text has been revised as suggested (lines 271-278)

The paper is very interesting describe some practical activity performed by the authors on site but the technical description is very poor. English review is needed.

An extensive revision of the English language and style was carried out.

Reviewer 2 Report

The paper presents relevant information on technical aspects of the reconstruction of small dams and their functionality. However, results related to the renewed presence of water on the surface and in the subsoil are scarce or even non-existent (subsoil). The authors report that monitoring is ongoing. The weakness of this paper is related to the little data related to monitoring (social and economic benefits and others). If more data can be included, the quality of the paper can improve significantly.

Author Response

The authors thank the reviewer for the careful review and for the suggestions received for the improvement of the paper. The responses to the review points are shown below:

The paper presents relevant information on technical aspects of the reconstruction of small dams and their functionality. However, results related to the renewed presence of water on the surface and in the subsoil are scarce or even non-existent (subsoil). The authors report that monitoring is ongoing. The weakness of this paper is related to the little data related to monitoring (social and economic benefits and others). If more data can be included, the quality of the paper can improve significantly.

All the data available to us based on the most recent survey and oral communications with the inhabitants has been added to the paper (lines 436-463). The current situation due to the COVID 19 Pandemic is preventing regular monitoring of further data.

Reviewer 3 Report

This manuscript is within scope of the journal. Water scarcity is a challenging problem that affects human life at several levels. The manuscript claims construction of two barrages to enhance water retention in the subsoils of the study area as a solution to the problem. The problem has been well described in the introduction section. Since the study is still in progress, any outcome from construction of barrages has not been tested yet. My comments on this manuscript are as follows:

The study projects renewed presence of water on the surface and in the subsoils after two barrages were built around Kita. This should be explicitly described at the end of the introduction section so that readers are not lost figuring out what to expect from this paper.  The introduction section is very broad and provides a lot of information. However, it appears to leave the reader stranded from the focus of this paper.

Layout of this manuscript should be fixed to use the space on the left side.

Figure 1 should also have a geographical coordinate reference for the map.

Figure 3: I don’t think it is a good idea to include a graph of number of people died and displaces during wars over the course of a certain period in a manuscript focusing on water regimentation. Number of people displaced is indirectly related to the theme of this paper but still the pattern of migration (70% ivory coast) does not warrant a graph in the introduction section.

Line 187: I see some information in the 2. materials and method section that are not much related to what this manuscript is about. E.g …developed collaborations specially in South America…I do not think this is much relevant in this section and looks like a filler.

Figure 4: Can we also demonstrate a conceptual diagram of underground barrage? It would be great if it could be demonstrated with the help of a diagram how surface and underground barrage are combined in a single structure (line-257)

Line-383 ‘to deliver’ instead of ‘for deliver’. There are some other typos in this manuscript that should be corrected.

Can we add a paragraph in the discussion section comparing the two barrages (Toumbouba and Kofeba). Why one could be more efficient than other in terms of investment (human resource, time, capital) and outcome (projected improvement in water retention etc.)

Line 430- “to the increase in mi- 430 cro-business activities related to the presence of water” This conclusion should not be made without supporting by data.

Line 435- “spread of the culture 435 of waste recycling, especially of an electronic nature, of which the consumer civilization 436 of the countries north of the equator” I don’t think this future direction is much related to the present study.

Author Response

The authors thank the reviewer for the careful review and for the suggestions received for the improvement of the paper. An extensive revision of the English language and style was carried out.

The responses to the review points are shown in red below:

The study projects renewed presence of water on the surface and in the subsoils after two barrages were built around Kita. This should be explicitly described at the end of the introduction section so that readers are not lost figuring out what to expect from this paper. The introduction section is very broad and provides a lot of information. However, it appears to leave the reader stranded from the focus of this paper.

The following sentence was added at the end of the introduction to better clarify the purpose of the paper (Lines 95, 96):

[…] The paper focuses on a research, design and implementation process aimed at the construction and upgrade of existing barrages in rural areas of Sub-Saharan Africa.

Layout of this manuscript should be fixed to use the space on the left side.

For the tables and figures that required it, the left side space was used (as suggested by the reviewer) respecting the template of the journal.

Figure 1 should also have a geographical coordinate reference for the map.

Geographic coordinates have been added on the figures 1 and 2.

Figure 3: I don’t think it is a good idea to include a graph of number of people died and displaces during wars over the course of a certain period in a manuscript focusing on water regimentation. Number of people displaced is indirectly related to the theme of this paper but still the pattern of migration (70% ivory coast) does not warrant a graph in the introduction section.

The mentioned graph has been deleted as suggested.

Line 187: I see some information in the 2. materials and method section that are not much related to what this manuscript is about. E.g …developed collaborations specially in South America…I do not think this is much relevant in this section and looks like a filler.

The superfluous information has been deleted from the paper.

Figure 4: Can we also demonstrate a conceptual diagram of underground barrage? It would be great if it could be demonstrated with the help of a diagram how surface and underground barrage are combined in a single structure (line-257)

The Concept diagram of the Underground barrages has been added to the figure 4.

Line-383 ‘to deliver’ instead of ‘for deliver’. There are some other typos in this manuscript that should be corrected.

The mistake has been changed as suggested by the reviewer. Moreover, an extensive revision of the English language and style was carried out in all the text.

Can we add a paragraph in the discussion section comparing the two barrages (Toumbouba and Kofeba). Why one could be more efficient than other in terms of investment (human resource, time, capital) and outcome (projected improvement in water retention etc.).

A comparison text between the two barrages has been added at the end of point 3.2 (from line 361) including a table for the comparison between the two structures.

Line 430- “to the increase in micro-business activities related to the presence of water” This conclusion should not be made without supporting by data.

All the data available to us based on the most recent survey and oral communications with the inhabitants has been added to the paper (lines 436-463). The current situation due to the COVID 19 Pandemic is preventing regular monitoring of further data.

Line 435- “spread of the culture 435 of waste recycling, especially of an electronic nature, of which the consumer civilization 436 of the countries north of the equator” I don’t think this future direction is much related to the present study.

The future directions have been removed.

Round 2

Reviewer 1 Report

My suggestion were followed. I agree.

Reviewer 2 Report

The authors made a great effort to improve the paper.